# A New Evidence Weight Combination and Probability Allocation Method in Multi-Sensor Data Fusion

**DOI:** 10.3390/s23020722

**Published:** 2023-01-08

**Authors:** Li Ma, Wenlong Yao, Xinguan Dai, Ronghao Jia

**Affiliations:** 1College of Communication and Information Engineering, Xi’an University of Science and Technology, Xi’an 710054, China; 2Xi’an Key Laboratory of Heterogeneous Network Convergence Communication, Xi’an 710054, China

**Keywords:** D-S evidence theory, maximum entropy, multi-sensor data fusion, the weight combination, trust discount

## Abstract

A single sensor is prone to decline recognition accuracy in the face of a complex environment, while the existing multi-sensor evidence theory fusion methods do not comprehensively consider the impact of evidence conflict and fuzziness. In this paper, a new evidence weight combination and probability allocation method is proposed, which calculated the degree of evidence fuzziness through the maximum entropy principle, and also considered the impact of evidence conflict on fusing results. The two impact factors were combined to calculate the trusted discount and reallocate the probability function. Finally, Dempster’s combination rule was used to fuse every piece of evidence. On this basis, experiments were first conducted to prove that the existing weight combination methods produce results contrary to common sense when handling high-conflicting and high-clarity evidence, and then comparative experiments were conducted to prove the effectiveness of the proposed evidence weight combination and probability allocation method. Moreover, it was verified, on the PAMAP2 data set, that the proposed method can obtain higher fusing accuracy and more reliable fusing results in all kinds of behavior recognition. Compared with the traditional methods and the existing improved methods, the weight allocation method proposed in this paper dynamically adjusts the weight of fuzziness and conflict in the fusing process and improves the fusing accuracy by about 3.3% and 1.7% respectively which solved the limitations of the existing weight combination methods.

## 1. Introduction

In recent years, human behavior recognition has received more attention because monitoring a person’s state provides valuable information for safety and health. The methods based on a visual sensor and an inertial sensor are the main research methods of human behavior recognition, but the methods based on vision have some disadvantages. The video data may be occluded, and further video recording will cause privacy problems in many cases, due to the illumination conditions, clothing types and background colors. Additionally, many sensors need infrastructure support, such as installing cameras in the monitoring area. Compared with other human behavior data acquisition equipment, accelerometers, gyroscopes, heart rate sensors, and other sensors with small sizes, and high-detection accuracy have been widely used in human behavior recognition.

However, the sensor itself is vulnerable to the environment, equipment aging, and interference of other impact factors in the signal acquisition process, resulting in abnormal data and uncertainty, a single sensor can no longer meet the demand for identification reliability. The multi-sensor data fusion technology proposed in the 1980s still has wide application prospects in this field. Up to now, some classical multi-sensor data fusion algorithms have been developed, mainly divided into four types: estimation theory, statistical inference, information theory, and artificial intelligence. The common data fusion algorithms include weighted average, Kalman filter, Bayesian estimation, evidence theory, neural network, and so on. In the face of the uncertainty generated by sensor abnormal data, evidence theory and Bayesian estimation are usually used. D-S evidence theory [1,2] can flexibly and effectively model uncertain information without prior probability, and use the base probability allocation function to reflect the probability of uncertain problems. Multi-sensor data fusion is divided into signal-level fusion, feature-level fusion, and decision-level fusion. D-S evidence theory is a decision-level data fusion algorithm. Decision-level fusion can take into account the advantages of different sensors, without being limited by the type or structure of sensor data, and make full use of the information provided by sensors. In addition, another feature of decision-level fusion is that the fusion information has a relatively small amount of calculation. So it can be flexibly used in multi-sensor classification problems with its strong capacity for heterogeneous sensors, such as fault diagnosis [3], target recognition [4,5], environment grade evaluation [6], etc.

Although D-S evidence theory can be applied in many fields, it may produce results contrary to common sense when handling highly uncertain evidence. Uncertain evidence consists of conflicting evidence and fuzzy evidence. Aiming at the impact of these two pieces of evidence on fusion, researchers have proposed many improvement methods to improve fusing accuracy. There are two ways to alleviate the above shortcomings in the combination rules of evidence theory. The first is to modify Dempster’s combination rule. Scholars who applied this method believed that the shortcomings of the original D-S evidence combination rule lead to the failure of handling conflicting evidence and the results of going against common sense. Therefore, Yager [7] defined conflict as an unknown region that exists objectively and assigned the conflict parts in evidence that could not be effectively determined by the D-S combination rule to this region, which actually discarded all the conflict parts of the evidence. By this method, the excellent mathematical properties of D-S evidence theory were lost. Therefore, most researchers chose the second method, that is, modification of the evidence source before the evidence combination, and the evidence with high credibility will obtain higher weight. The credibility of the evidence is measured by the conflict degree between pieces of evidence and its fuzzy degree. Sun, Y [8] assigned different weights to each sensor according to the different characteristics of the two sensors. Yang, K [9] combined the Jousselme distance between pieces of evidence with the overall evidence decision to comprehensively measure the evidence conflict. Qiang, C [3] proposed a new correlation coefficient that has a better performance in measuring the relationship between quality functions. Si, L [10] introduced the correlation coefficient of evidence to measure the conflict between pieces of evidence in order to enhance the fusing effect and improve the applicability, as well as adopting the weighted average combination model to modify and preprocess the original evidence. Some researchers used information entropy to measure evidence fuzziness. For example, H¨ohle entropy [11], Deng entropy [12], Yager’s [13] dissonance measurement, and Jousselme [14] defined an entropy function based on the idea of Pignistic transformation to measure the fuzziness in evidence theory. Among them, Deng entropy is the most widely used one. Zheng, H [15] used Deng entropy to measure the uncertainty of expert evaluation and then determined the weight of each risk factor. Yan, H [16] fully considered the relationship between subsets based on Deng entropy and proposed an improved entropy, which has greater advantages in the measurement of evidence fuzziness.

It can be seen from the above analysis that researchers have conducted in-depth research on the measurement methods of evidence conflict and evidence fuzziness, and made a lot of achievements, which have laid a solid foundation for the application of evidence theory. Yet most researchers tended to only consider conflict or fuzziness. Although some researchers have given comprehensive consideration to the two impact factors, they usually believed that fuzziness and conflict have the same impact on the fusing results.

Gou, L [17] proposed a multi-source data fusion model based on Euclidean distance and weight allocation. Specifically, Euclidean distance between evidence was calculated to reflect the similarity of evidence, and the similarity was used to assign different weights to each piece of evidence, and the initial BPA was modified before fusion to reduce the impact of evidence conflict on fusion results. Experiments show that this method can improve the accuracy of the model and has certain advantages in the accuracy and timeliness of fault diagnosis. Similarly, Hua, Y [18] proposed the concept of compatibility ratio to measure the degree of conflict among pieces of evidence. By calculating the compatibility coefficient between a certain piece of evidence and the expected value of all evidence, the credibility of the evidence can be calculated. The evidence is then given a weight that can be used to adjust BPA to overcome problems caused by conflicting evidence. However, the author only considers the impact of conflict on the fusion results, and indeed ignores the degree of fuzziness of evidence. 

Jing, M [19] assigned the basis confidence function to each mutex according to classical probability theory, and calculated the arithmetic mean to modify the initial BPA. It can be understood that the same initial probability is assigned to each subset in the recognition framework, and this initial prior probability is used to overcome the shortcomings of D-S evidence theory in the fusion. This method is a consideration of the degree of fuzziness of evidence, equivalent to assigning the same degree of fuzziness to each piece of evidence, which is similar to the maximum entropy theory mentioned later in this paper, but this paper uses this average distribution to calculate the degree of fuzziness of evidence, rather than directly taking this average distribution as the degree of fuzziness of evidence.

Wang, J [4] analyzed the existing methods and found that in the process of weight allocation of evidence, most scholars focus on the conflict measurement of evidence while ignoring or underestimating the influence of evidence fuzziness on fusion results. Therefore, the current weight calculation mainly depends on the degree of conflict. Then it is proved that the fuzzy degree of evidence has a significant influence on the fusion result. Additionally, on the basis of evidence credibility (degree of conflict) and clarity (degree of fuzziness). A new weight allocation method is proposed, which takes the credibility and clarity of evidence as the weight of evidence. Finally, the effectiveness and superiority of this method are proved by experiments.

However, in some real cases, the above simple weight allocation combination will eventually lead to a large deviation between the fusion result and the real result. As shown in Experiment 1 in Section 4, due to the unreasonable weight allocation, the fuzzy factor obtains too high a weight, which leads to errors in the final fusing results. Therefore, for the shortcomings of the above combination methods, a new weight combination and probability allocation method is proposed. The main contributions are as follows.

On the basis of measuring the clarity and conflict degree of evidence correctly, a new weight combination method is proposed to combine the two influence factors, which makes the weight distribution of fuzziness and conflict more reasonable. The combined weight is used to calculate the discount coefficient, and the basic probability distribution is redistributed to reduce the influence of evidence conflict and fuzzy on the fusion result. Finally, the proposed method is applied to human behavior recognition, and the results show that the proposed method can effectively resolve the influence of conflicting evidence and fuzzy evidence on fusion results.

The rest of the paper is as follows: Section 2 introduces the conflict and fuzzy measures. In Section 3, the weight combination, and a new allocation method of probability function based on trust discount are introduced. In Section 4, the limitations of the existing methods are demonstrated by experiments. Then, the proposed method is verified by using the PAMAP2 behavior dataset and compared with several other methods, which proves the advantages of the proposed method. Additionally, introduces the application of this method in human behavior recognition. The last section is a conclusion.

## 2. Preliminaries

### 2.1. Dempster-Shafer Evidence Theory

The evidence theory is established by the famous scholar Shafer based on Dempster’s combination rule, so it is also called D-S evidence theory. Compared with other data fusion algorithms, D-S evidence theory has higher efficiency in solving uncertainty problems. It is mainly viewed and analyzed by transforming propositions into mathematical sets. Since a set can contain multiple elements, different from probability theory, which only considers a single element, it is because of the fuzziness of evidence theory that it can better express the uncertainty of propositions. In fact, it is more like a simulation of the human brain’s integrated processing of events under the influence of multiple environmental factors. The first problem is to observe and collect information, namely evidence. Then the information from all aspects is integrated to make judgments, and the final results of the problem are obtained, that is, evidence combination. The most important ones are to determine the range of answers (recognition framework), the probability corresponding to evidence set assignment (base trust assignment function), and the combination of evidence probability data (Dempster’s combination rule).

**Definition 1.** *Recognition framework: If all elements in a complete set Θ are mutually exclusive, and the number of elements is enumerable and finite, the set can be called a recognition framework, denoted as* [1,2]:(1)$$\Theta =\left\{\theta_{1}\;,\theta_{2},\cdot \cdot \cdot \;,\theta_{n}\right\}$$

**Definition 2.** *Base trust assignment function: The base trust assignment function m on the recognition framework Θ is a mapping from the set*$2^{\Theta }\to \left[0,\;1\right]$*, and satisfies* [1,2]
(2)$$\{\;\begin{array}{l} m\left(\emptyset \right)=0\\ \underset{A\subseteq \Theta }{\sum }m\left(A\right)=1 \end{array}$$

**Definition 3.** *Dempster’s combination rule: Assuming that the base trust assignment function corresponding to two pieces of evidence is*$m_{1}$*and*$m_{2}$*in the recognition framework Θ, Dempster’s combination rule is* [1,2]:(3)$$\{\begin{array}{l} m\left(\emptyset \right)=0\\ m\left(A\right)=\frac{1}{1-k}\underset{A=A_{i}\cap B_{j}}{\sum }m_{1}\left(A_{i}\right)m_{2}\left(B_{j}\right) \end{array}$$
*where*
k
* is the conflict coefficient, which is defined as follows:*
(4)$$k=\sum_{A_{i}\cap B_{j}=\emptyset }m_{1}\left(A_{i}\right)m_{2}\left(B_{j}\right)$$

### 2.2. Theory of Maximum Entropy

The essence of the maximum entropy principle is that when we need to predict the probability distribution of a random event, we should not make any subjective assumption about the unknown situation under the premise of partial knowledge. In this case, the probability allocation is the most uniform and the risk of prediction is the least. At this time, the information entropy of the probability allocation is the maximum. The advantage of meeting the maximum entropy distribution is that the result can cover the currently known feasibility without making any subjective guess. Such a model is the smoothest and the risk of prediction is the least. When faced with uncertainty, we should keep all the possibilities open instead of making arbitrary assumptions. Jing, M [19] improved the assignment method of BPA (base probability allocation) by applying the maximum entropy theory to assign the same confidence to each basic mutex of evidence. This prior probability, to a certain extent, overcame the deficiency of Dempster’s combination rule in fusing conflicting data. The theory of maximum entropy can also be used to measure the uncertainty of evidence. According to the maximum entropy principle, when BPA satisfies uniform distribution, the information entropy reaches the maximum, and the uncertainty also reaches the maximum. Wang, J [4] measured the degree of evidence fuzziness by calculating the distance between BPA and uniform distribution. The uniform distribution is defined as follows [4]:(5)$$Unif\left(E_{i}\right)=\left(\frac{1}{\left|E_{i}\right|},\frac{1}{\left|E_{i}\right|},\ldots ,\frac{1}{\left|E_{i}\right|}\right)$$
where $\left|E_{i}\right|$ represents the number of focal elements (the number of elements in the recognition framework Θ) in evidence $E_{i}$.

### 2.3. Conflict Measures of Evidence

When measuring evidence conflict, the most commonly used method is to measure conflict by the distance between pieces of evidence. Jousselme distance [20] is an effective measurement method of evidence conflict. The following are some introductions:

Assuming that there are two pieces of evidence $E_{1}$ and $E_{2}$ and their basic trust assignment functions $m_{1}$ and $m_{2}$ under the recognition framework Θ, the focal elements are $A_{i}$ and $B_{j}$, respectively, and $m_{1}$ and $m_{2}$ are regarded as row vectors, the Jousselme distance between $m_{1}$ and $m_{2}$ can be expressed as follows [20].
(6)$$J_{BPA}\left(m_{1},m_{2}\right)=\sqrt{\frac{1}{2}\left(m_{1},m_{2}\right)D{\left(m_{1},m_{2}\right)}^{T}}$$
where D is $2^{N}\times 2^{N}$ symmetric matrix (N=|Θ|), matrix elements are:(7)$$d_{ij}=\frac{|A_{i}\cap B_{j}|}{|A_{i}\cup B_{j}|}$$

After obtaining the distance between evidences, the similarity of evidence body $m_{i}$ is defined as:(8)$$\mathrm{Sim}\left(m_{i}\right)=\sum_{j=1,i\neq j}^{n}\left(1-J_{BPA}\left(m_{i},m_{j}\right)\right)$$

In addition, the conflict can be measured from a geometric perspective. In mathematics, the cosine of an angle can measure the difference between the directions of two vectors. Similarly, in D-S evidence theory, this concept can also be used to measure the similarity between evidence vectors.

Assuming that evidence $m_{i}$ and $m_{j}$ are mutually independent in the recognition framework Θ, the cosine similarity of evidence $m_{i}$ and the evidence $m_{j}$ is defined as follows [21]:(9)$$Sim_{ij}=\cos_{m_{i}m_{j}}=\frac{\langle m_{i}\cdot m_{j}\rangle }{\left|m_{i}\right|\cdot \left|m_{j}\right|}$$
where $\langle m_{i}\cdot m_{j}\rangle$ represents the product of vector $m_{i}$ and $m_{j}$, and $|m_{i}|$ represents the modulus of the vector $m_{i}$.

Ren [21] combined the above evidence distance and evidence angle and proposed a measurement method of conflict that is to measure the distance and angle difference between pieces of evidence from vector and geometric perspectives.

This paper mainly studies the combination method of conflict and fuzzy measures, so the widely used Jousselme distance is chosen to measure conflict.

### 2.4. Fuzzy Measures of Evidence

Information entropy is a classical method to measure the uncertainty of evidence. Shannon entropy has a good performance in measuring the uncertainty of evidence. Shannon entropy can be used to calculate the amount of information contained in the evidence body, and then calculate the uncertainty of evidence. Höhle [11] first defined the entropy of BPA in evidence theory as:(10)$$H\left(m\right)=-\sum_{A\subseteq \Theta }m\left(A\right)\log_{2}Ble\left(A\right)\;\;\;$$
where m(A) is a BPA in the recognition framework Θ and Ble(A) is a belief function in Θ.

On this basis, Deng [12] proposed Deng entropy, which is defined as:(11)$$D\left(m\right)=-\sum_{A\subseteq \Theta }m\left(A\right)\log_{2}\frac{m\left(A\right)}{2^{\left|A\right|}-1}\;\;\;\;\;\;$$
where m is the BPA defined on the recognition framework Θ, and |A| is the number of focal elements of A.

Due to the strong ability to measure the uncertainty of BPA, Deng entropy has attracted the attention of many researchers [22,23] and has been improved from multiple perspectives [24,25,26], which has been widely used in the field of multi-sensor data fusion [27].

Other scholars measured the fuzziness of evidence from the perspective of probability distribution [4]. According to the maximum entropy principle, the probability of occurrence of events in the system satisfies all known constraints, and no assumption is made on any unknown information, that is, for unknown, it is treated as an equal probability. When all focal elements in evidence have an equal probability distribution, the uncertainty of evidence reaches the highest, that is, the highest degree of fuzziness. Qin, Y [28] proposed a distance-based uncertainty measurement method on the basis of existing studies, which determined the total uncertainty measure by calculating the Euclidean distance between the monad set and the least uncertain interval. In addition, the degree of evidence fuzziness can be estimated by calculating the Wasserstein [29,30] distance between BPA and uniform distribution [4]. The smaller the distance, the higher the degree of fuzziness. Wasserstein distance measures the distance between two variables according to the probability distribution [31,32]. Its biggest advantage lies in that even if the support sets of two distributions do not overlap or overlap very little, they can still reflect the distance between the two distributions. Therefore, in this paper, Wasserstein distance is used to prove the degree of fuzziness. Wasserstein distance is defined as follows:

Assuming that we will transform the probability distribution p(x) into q(x) and set the distance function (transfer cost) as d(x,y), then Wasserstein distance is defined as [31,32]:(12)$$Wass\left(p,q\right)=\underset{\gamma \in \Pi \left[p,q\right]}{\inf}\iint \gamma \left(x,y\right)d\left(x,y\right)dxdy$$

γ∈Π[p,q] is the joint distribution of p,q.

The Wasserstein distance between the evidence $E_{i},E_{j}$ under the recognition framework Θ$=\left\{\theta_{1},{\;\theta }_{2},\cdots ,{\;\theta }_{n}\right\}$ is as follows [31,32].
(13)$$Wass\left(E_{i},E_{j}\right)=\underset{\gamma \sim \Pi \left[E_{i},E_{j}\right]}{\inf}{Ε}_{\left(x,y\right)\sim \gamma }\left[\left\|x-y\right\|\right]$$
where $\gamma \sim \Pi \left[E_{i},E_{j}\right]$ is the set of all possible joint distributions combined by the distributions of pieces of evidence $E_{i}$ and $E_{j}$. For every possible combination distribution γ, we can obtain a sample x and y from sampling (x,y)~γ and calculate the distance of the sample ‖x−y‖. So under the joint distribution of γ, the sample’s expected value of the distance E(x,y)~γ[‖x−y‖] can be calculated. The lower bound $inf\gamma \sim \Pi \left[E_{i},E_{j}\right]$ E(x,y)~γ[||x−y||] that we can take on this expected value in all possible joint distributions is the Wasserstein distance. 

By calculating the Wasserstein distance between the BPA and uniform distribution, the clarity of evidence can be obtained, which is defined as:(14)$$Clar\left(m_{i}\right)=Wass\left(Unif(E_{i}),BPA\right)$$

### 2.5. Existing Problems

In this section, common examples are used to illustrate the counterintuitive problems of the existing weight combination methods in handling high-conflict and high-clarity evidence data fusion of the combination rules of D-S evidence theory.

Supposing a recognition framework is given as Θ={A,B}, $m_{1},m_{2}\;and\;m_{3}$ are the three pieces of evidence under the framework. The evidence bodies $m_{1}$ and $m_{2}$ have the highest support for proposition B, and $m_{3}$ has the higher support for proposition A. The degree of conflict between $m_{1}$ and $m_{2}$ is relatively small, but the degree of fuzziness is higher than that of $m_{3}$. Specific values are shown in Table 1:

On the basis of existing research, considering the measurement of evidence conflict (Equation (8)) and fuzziness (Equation (4)) comprehensively, the existing methods are used to directly multiply the weight coefficients of conflict and fuzziness or assign the same weight (arithmetic average) to conflict and fuzziness. The fusing results under the Dempster combination rule are shown in Table 2 (specific steps can refer to experiment 1):

According to the fusing results in Table 2, under Dempster’s combination rule, even if the pieces of evidence $m_{1}$ and $m_{2}$ pointed to the correct result m(B) at the same time, because the evidence $m_{3}$ had higher clarity, according to the existing weight combination methods, the evidence $m_{3}$ got too high weight, so that the final fusing result pointed to the proposition m(A). Therefore, the results of the existing methods in the fusion of high-conflict and highly-clarity pieces of evidence are not accurate and even wrong.

## 3. A New Method of Weight Allocation

Based on the effective measurement of evidence conflict and evidence fuzziness, this Section analyzed the influence of weight combination rules on fusing results and proposed a new adaptive weight allocation method, which effectively solved the limitations of existing methods. On this basis, a multi-sensor data fusion model was established, as shown in Figure 1:

Firstly, the distance between each sensor and the corresponding grade interval was calculated to generate BPA. Then, the degree of evidence fuzziness was calculated according to the method in Wang [4]. At the same time, Jousselme distance was used to calculate the distance between pieces of evidence to reflect the degree of evidence conflict. On this basis, a new combination method was used to determine the final weight of evidence. The BPA was redistributed according to the trust discount. Finally, the pieces of evidence were fused according to Dempster’s combination rule to obtain the decision results.

### 3.1. New Weight Combination Method

As can be seen from the examples in Section 2.5, under the conditions of high conflict and high clarity, the results of traditional methods will be biased in favor of evidence with higher clarity. After considering the conflict, the results will naturally be biased in favor of evidence with a lower conflict degree. The influence of fuzzy on fusion results is also very significant, it can make the results obviously favor the high-clarity evidence. However, although the existing weight combination scheme comprehensively considers the impact of conflict and fuzziness on the fusion results, in the face of the high degree of evidence conflict and clarity, high-clarity evidence will gain too much weight, and eventually, lead to wrong decision results. Therefore, a more reasonable weight allocation scheme is needed, which can automatically adjust the conflicting and fuzzy weights under some extreme conditions, so that the results can be closer to the real situation. On the basis of correctly measuring the fuzziness and conflict of evidence, we propose a new adaptive weight combination formula:(15)$$W\left(m_{i}\right)=\frac{1}{2}\left(Sim\left(m_{i}\right)+\frac{Clar\left(m_{i}\right)}{e^{\left|\left(1-Sim\left(m_{i}\right)\right)+Clar\left(m_{i}\right)-1\right|}}\right)$$
where $Clar\left(m_{i}\right)$ represents the clarity of evidence, $Sim\left(m_{i}\right)$ represents the similarity of evidence, and $W\left(m_{i}\right)$ is the weight of evidence $m_{i}$. With the characteristic of the rapid rise of the index of e when it is greater than zero when the evidence points to the wrong results because of its problems or being interfered with by other factors, it cannot obtain too high weight even with high clarity, so as to reduce the adverse impact on the fusing results.

### 3.2. Probabilistic Reallocation Method Based on Trust Discount

On the basis of obtaining the weight of evidence (trust degree), BPA is modified by the trust discount formula, denoted as $\mathrm{NewBpa}\left(m_{i}\right)$:(16)$$\mathrm{NewBpa}\left(m_{i}\right)=\mathrm{Bpa}\left(m_{i}\right)*W\left(m_{i}\right)+{\left(1-W\left(m_{i}\right)\right)}^{2}*\left(1-\mathrm{Bpa}\left(m_{i}\right)\right)$$
where $\mathrm{Bpa}\left(m_{i}\right)$ represents the base probability allocation, and $\mathrm{NewBpa}\left(m_{i}\right)$ represents the probability distribution after reallocation according to the weight of evidence $W\left(m_{i}\right)$.

## 4. Application in Activity Recognition

### 4.1. Introduction to the DATA Set

The PAMAP2 (physical activity monitoring data set) contains 18 different physical activity data (such as standing, walking, running, etc.) performed by nine subjects wearing three inertial measurement devices and heart rate monitors. This data set can be used for activity recognition and intensity estimation, as well as developed and applied to data processing, segmentation, feature extraction, and classification algorithms. In order to justify the research method, we selected five actions (lying, standing, walking, going up and down stairs, and running) from the PAMAP2 data set and used the proposed method for behavioral classification.

The main focus of this study is on the measurement of conflict and fuzziness as well as weight allocation. Therefore, the feature extraction process of the original data is simplified. To classify these 18 actions, we are required to deal with the original data and choose appropriate methods for feature extraction. In order to reduce the test period and complexity, only five actions (lying, standing, walking, going up and down stairs, and running) were selected in the PAMAP2 data set for classification.

### 4.2. Experiment 1

The multi-sensor data fusion model proposed in Section 3 is applied to behavior recognition. Let the recognition framework be Θ={A,B,C,D,E}, corresponding to lying, standing, walking up stairs, and running, respectively. The acceleration value (ACC), angular velocity value (GYRO), and heart rate value (BPM) collected by the sensor were processed and fused for behavior classification. The arithmetic average value of sensors corresponding to each action is taken as the characteristic value of behavior parameters, as shown in Table 3.

A set of sensor data with high conflict and clarity in walking was selected for the experiment, and the data collected by each sensor was S=( 9.7728, 0.39, 101).

Step 1: Calculate the Euclidean distance between each sensor data and the corresponding behavioral feature level, and then normalize it as BPA. As shown in Table 4.

Step 2: Calculate Jouselme distance between pieces of evidence according to Equations (6) and (7), and then calculate the similarity of evidence bodies (Equation (8)). $Sim\left(m_{1}\right)=$0.563, $Sim\left(m_{2}\right)=$0.743, $Sim\left(m_{3}\right)=$ 0.694.

Step 3: Calculate the Wasserstein distance between each piece of evidence and uniform distribution (5) according to formula (14), and the evidence clarity can be obtained after normalization. $Clar\left(m_{1}\right)=0.551,\;Clar\left(m_{2}\right)=0.252,\;Clar\left(m_{3}\right)=0.197$.

Step 4: Calculate the weight of the evidence. The evidence similarity and clarity calculated in steps 2 and 3 are taken into the weight combination formula (15) proposed in Section 3 for calculation, and the following results can be obtained: $W\left(m_{1}\right)=0.414$, $W\left(m_{2}\right)=0.730$, $W\left(m_{3}\right)=0.607$.

Step 5: Reallocate BPA. Put the weight calculated in Step 4 into Equation (16) to reallocate BPA.

Step 6: Combing the rules (3) and (4) according to D-S evidence theory, the modified BPA is fused to obtain the fusing results of multi-sensor (evidence) data, and then compare with other methods, as shown in Table 5 and Figure 2.

It can be seen from the experimental results, both the traditional method of D-S evidence theory (blue) and the proposed method (gray) can obtain the correct classification results. Wang, J’s [4] method (orange) considered the fuzziness’s influence on fusing results but ignored the influence of different weights of fuzziness and conflict on the fusing result, which caused the final result to be incorrect. The method proposed in this paper made the result closer to the real situation while considering the conflict and fuzziness.

### 4.3. Experiment 2

In order to verify the recognition effect of the method under different conditions, this experiment classifies different behaviors in the PAMAP2 data set (50%~100% of the data are randomly selected from the total sample) and calculates the accuracy of behavior recognition to verify the applicability of the method proposed in this paper and make a comparison with other methods. The results are shown in Table 6. Figure 3 and Figure 4 show the recognition accuracy of different methods under different numbers of sensors.

The results in Table 6 show that compared with the traditional D-S evidence theory and the improved method proposed by Wang, J [4], the proposed method in this paper is improved by 3.1% and 1.6% respectively.

In order to better evaluate the proposed multi-classification model, we choose the Kappa coefficient, which is commonly used in multi-classification problems, to evaluate our model. The Kappa coefficient is calculated as follows:(17)$$K=\frac{P_{0}-P_{e}}{1-P_{e}}$$
where $P_{0}$ represents the overall classification accuracy, and $P_{e}$ represents the sum of the products of the actual and predicted numbers corresponding to all classes divided by the square of the number of samples. The evaluation criteria are shown in Table 7:

By calculation, the kappa coefficient of the multi-classification model proposed in this paper is 0.8357.

It can be seen from the experimental results that the conflict and fuzziness of evidence have different degrees of influence on fusion accuracy. The proposed method distributes the weight of conflict and fuzziness more reasonably and improves the fusion accuracy, which can be seen from the following three points.

(1) Conflict and Fuzzy represent the measurement of conflict and fuzziness of evidence respectively. Compared with the traditional method (D-S), these two methods calculate the degree of conflict and fuzziness of evidence, respectively, and reallocate BPA, which improves the fusing accuracy by about 1.3%. Wang, J [4] used the multiplication of conflict and fuzziness coefficients as weights to reallocate BPA, which improved the fusing results by about 1.7% compared with the traditional method, indicating that comprehensively considering the conflict and fuzziness of evidence is an effective method to improve the fusing accuracy. Based on the existing research, this paper (red line) dynamically allocates the weight coefficients of conflict and fuzziness, so that the fusing results are about 3.1 % higher than the traditional method, and about 1.6% higher than the Wang, J ’s [4] method, showing that the weight allocation method proposed in this paper is more reasonable.

(2) Comparing the results of Figure 3 and Figure 4, it can be seen that the proposed method can obtain better classification results under different numbers of sensors. Additionally, the improvement under three sensors is about 1.8% higher than that under two sensors, which shows that the proposed method can effectively solve the classification problem under multiple sensors.

(3) The results of the model evaluation show that the multi-sensor data fusion model proposed in this paper has good balance, and has reached a high level for the classification of different behaviors, which can be effectively applied to human behavior recognition.

The proposed weight allocation method dynamically adjusts the weight coefficients of conflict and fuzziness in the fusion process, and the fusion accuracy is improved by 3.1% and 1.6%, respectively, compared with the traditional method and the existing method. Moreover, high fusion accuracy and balance can be obtained under different sensor numbers. The problem of evidence conflict and fuzziness in the process of multi-sensor data fusion is solved effectively.

## 5. Conclusions

D-S evidence theory has been widely used in various fields since it was proposed, especially in dealing with decision problems under multi-sensor conditions. The method proposed in this paper aims to present a fusion method for embedded devices to solve the problems of data conflict and fuzziness in existing multi-sensor data fusion.

Based on the existing research, this paper selects a suitable method to measure the degree of conflict and fuzziness of evidence. A new weight combination scheme is proposed which can dynamically adjust the proportion of conflict and fuzziness in the fusion process. The experimental results show that the proposed method reduces the adverse effects of evidence conflict and is fuzzy on the results of data fusion, and the fusion accuracy of the model is improved by 3.1% and 1.6%, respectively, compared with the traditional method and the existing method. It has certain advantages in the accuracy of behavior classification problems.

The method proposed in this paper is not only suitable for human behavior recognition but also suitable for other systems which need to consider multiple factors. The focus of this study is data fusion, thus simplifying data preprocessing and feature extraction. Considering the complexity and uncertainty of classification problems, we will try more appropriate feature extraction methods in future work to further improve the fusion accuracy. In addition, this study also focuses on the fusion of single-mode data. In the future, we will try to fuse cross-modal data to obtain more reliable recognition results.

## Figures and Tables

**Figure 1 sensors-23-00722-f001:**
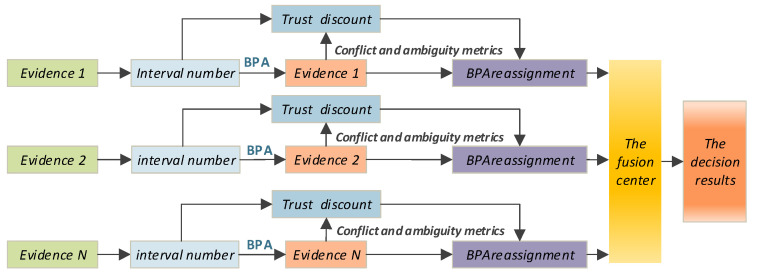
Multi-sensor data fusion model.

**Figure 2 sensors-23-00722-f002:**
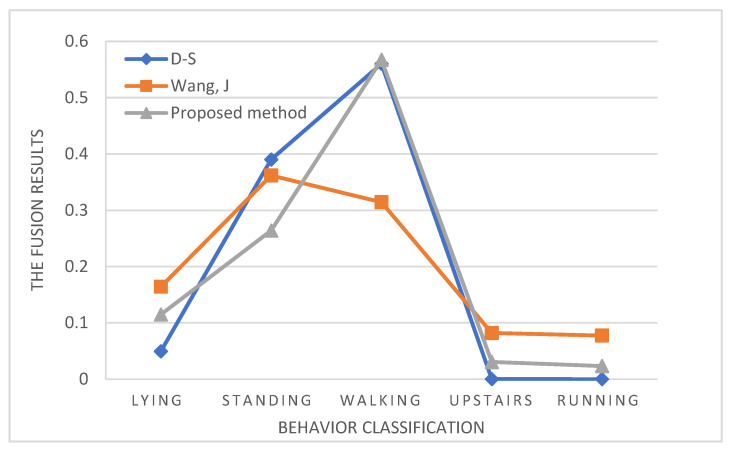
Fusion results of different methods [4].

**Figure 3 sensors-23-00722-f003:**
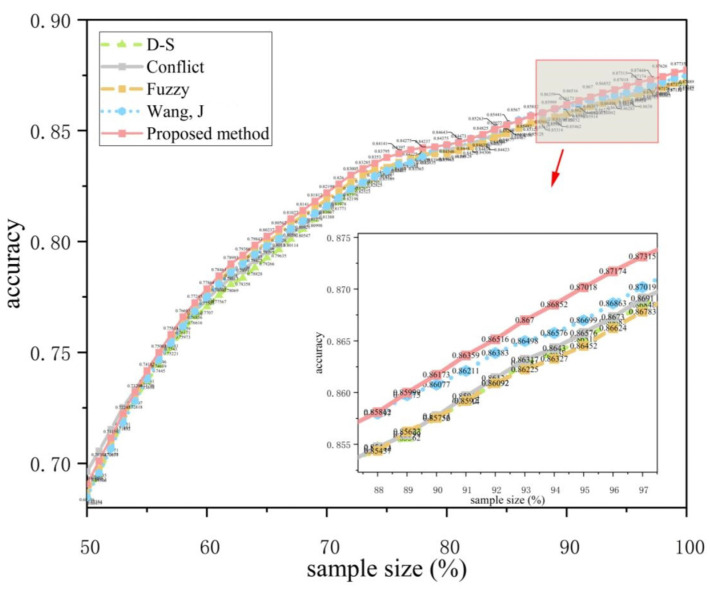
Accuracy of different methods with two sensors [4].

**Figure 4 sensors-23-00722-f004:**
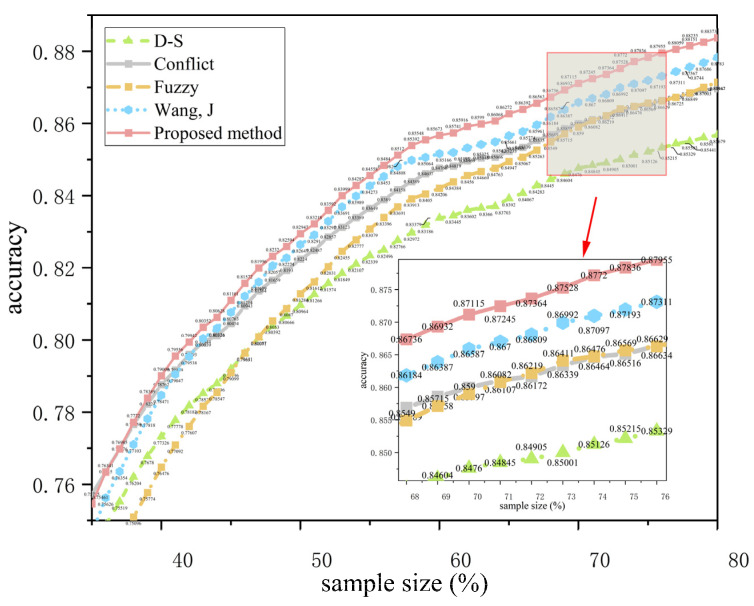
Accuracy of different methods with three sensors [4].

**Table 1 sensors-23-00722-t001:** Examples of high-conflict and high-clarity evidence.

Focus Element	m(A)	m(B)
$m_{1}$	0.3	0.7
$m_{2}$	0.3	0.7
$m_{3}$	0.9	0.1

**Table 2 sensors-23-00722-t002:** Fusion results of different methods.

Methods	m(A)	m(B)
D-S	0.6231	0.3769
Conflict	0.4101	0.5899
Fuzzy	0.7354	0.2646
Wang (multiply)	0.5841	0.4159
Jing (arithmetic mean)	0.5716	0.4284

**Table 3 sensors-23-00722-t003:** Behavior parameter feature table.

Focus Element	Lying	Standing	Walking	Upstairs	Running
ACC (m/$s^{2}$)	9.7987	9.7676	9.7433	10.8592	17.5110
GYRO (rad/s)	0.1868	0.2126	0.3021	2.8186	3.8023
BPM (PM)	87.5275	91.183	103.4257	120.4410609	161.3601

**Table 4 sensors-23-00722-t004:** BPA for each behavior parameter.

Focus Element	Lying	Standing	Walking	Upstairs	Running
ACC	0.145	0.724	0.127	0.003	0.001
GYRO	0.217	0.249	0.502	0.018	0.013
BPM	0.113	0.155	0.628	0.078	0.025

**Table 5 sensors-23-00722-t005:** Comparison of fusion results.

Methods	Lying	Standing	Walking	Upstairs	Running
D-S	0.0496	0.3901	0.5602	0.0001	0.0000
Wang, J [4]	0.1642	0.3619	0.3144	0.0821	0.0774
Proposed method	0.1148	0.2639	0.5674	0.0305	0.0234

**Table 6 sensors-23-00722-t006:** Accuracy of different methods.

**Focus Element**	**Lying**	**Standing**	**Walking**	**Upstairs**	**Running**	**Weighting Average**
**Sample size**	15,882	27,180	22,185	23,539	20,999	
D-S	0.9056	1	0.5597	0.9011	0.7875	0.8355
Wang, J [4]	0.9014	0.9979	0.7107	0.8	0.8211	0.8497
Proposed method	0.9016	0.9990	0.6410	0.8658	0.9066	0.8663

**Table 7 sensors-23-00722-t007:** Comparison of fusion results.

Coefficient	0~0.2	0.2~0.4	0.4~0.6	0.6~0.8	0.8~1
**consistency**	slight	fair	moderate	substantial	almost perfect

## Data Availability

The data used to support the findings of this study are available from the corresponding author upon request.

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
