# Peer review of "A New Evidence Weight Combination and Probability Allocation Method in Multi-Sensor Data Fusion"

_sensors, 2023, doi:10.3390/s23020722_

Round 1
Reviewer 1 Report (New Reviewer)
1. The related works section is missing, the authors must list the research related to their work.
2. The references list must be improved, and the authors should cite up-to-date references.
3. The keywords must be listed in alphabetical order.
4. The authors should highlight the contributions at the end of the introduction before the paper structure.
5. The equations should be referred to in the text.
Author Response
Please see the attachment.

Reviewer 2 Report (New Reviewer)
The introduction section is vague and confusing. Make it precise and more apparent.
Authors should elaborate on parameter importance analysis while deciding the methodology.
How to ensure the robustness of the proposed model?
Are the experiments performed on their own? Where is the setup figure? How were sensors selected, located, and data acquired?
In Table 3, what is the unit of acceleration and velocity?
How was diversity considered, like age, gender, psychological mindset, etc.?
Why was Euclidean distance preferred over other measures?
In Table 5, what do the numbers indicate? Fusion results of what?
Results and discussion are difficult to understand. What inferences have been drawn?
The conclusion should be revised according to achievements.
Round 2
Reviewer 2 Report (New Reviewer)
All my comments have been addressed carefully.
This manuscript is a resubmission of an earlier submission. The following is a list of the peer review reports and author responses from that submission.
Round 1
Reviewer 1 Report
In this paper, the authors proposed a new approach to combine bba in the framework of belief function theory. This combination is based on the calculation of weights which are used to discount initial bbas. These weights are computed with the degree of evidence fuzziness through the maximum entropy and with the impact of evidence conflict on fusing results. An experimental study, on one dataset called PAMAP2, is provided.
Even if the bibliography seems complete, despite the number of works in this domain, this paper is difficult to read. It could be improved (or corrected) by:
- rewritten introduction. The authors should make a better synthesis of works close to their proposal,
- section 2.2: I do not understand its interest: is it used to calculate the bbas or to determine a degree of uncertainty of the bba
- section 2.3: the authors use distance as a measure of conflict but some authors like [1] do not agree with this point even if more recently others have made the link between distance and conflict [2]. This point should be addressed.
- Several terms are used but they are not defined in the article: focal element, the notations of equation (9), $Ble$ in equation (10), the term $Clar$ in equation (14),...
- the end of section 2.4 is very difficult to understand. The link is also complicated to do with the rest of the article.
- the bbas presented in section 2.5 are very particular because they are Bayesian which makes the example a bit biased. Moreover, I do not understand the interest of this example because it is not taken up later with the proposed approach.
- Section 4.1: Why reduce the number of assumptions to 5? Authors should justify this choice.
- Line 285: the presentation of the values of $S$ should be in the same order as that of the table (i.e. ACC, GYRO and BPM and not BPM, ACC and GYRO).
- Table 4 is not the correct one (it is identical to table 3). This complicates the understanding of the example.
Reference:
[1] S. Destercke, T. Burger, Toward an axiomatic definition of conflict between belief functions, IEEE Trans. Syst. Man Cybern. B, vol.43, n°2, pp. 585–596, 2013.
[2] F. Pichon, A.-L. Jousselme, N. Ben Abdallah. Several shades of conflict. Fuzzy Sets and Systems, 366, pp. 63-84, 2019.
Reviewer 2 Report
The authors propose a new evidence weight combination and probability allocation method for Multi-sensor Data Fusion which calculates the degree of evidence fuzziness through the maximum entropy principle, and also considered the impact of evidence conflict on fusing results. The manuscript is well-written and has a technical content, however, authors need to consider the following suggestions/corrections.
1. What is effect and significance of existing problems with respect to counterintuitive problems of the existing weight combination methods?
2. In abstract authors just present the superiority of proposed method without quantified results (e.g., accuracy).
3. Concrete motivation with an example is missing at the beginning of the manuscript which emphasise the need for a novel method.
4. “Other scholars measured the fuzziness of evidence from the perspective of the probability distribution” – Line 183-184: no citation present.
5. “In this chapter, based on the effective measurement of evidence conflict and evidence fuzziness, the impact of the weight combination rule on fusing results was analyzed, and a new adaptive weight allocation method was proposed, which effectively solved the limitations of the existing methods. – Restructure the sentence.
6. Authors have not evaluated the new Multi-sensor data fusion model against the well define criteria.
7. Authors have not conducted analytical experimentation to prove the novelty and richness of the proposed method.
8. Experiments are conducted only for the trivial data sets in terms of size/features (only two applications are considered). Moreover, no analysis on the results and observations with justification for the results.
9. In conclusion should also summarise, the justification for receiving better results with respect to the state of the art along with quantified results in terms of improvement.
Reviewer 3 Report
The author needs to significantly proof read the paper before inclusion in the final version.
Why the author has advised to implement Dempster’s combination rule whereas other rules are available, authors are advised to discuss the reasonable evidence which suggests to use the above method.
Round 2
Reviewer 1 Report
I am not convinced by the author's answers. For example, for the example of Table 1. It is possible to present the problem with a more general numerical example (if the problem encountered only exists in this particular case it then becomes much less interesting). Moreover, the other experiments (experiments 1 and 2) are based on the same principle.
Also, I don't understand the values ​​obtained in table 4. For example, the computation of the euclidean distance between S and feature (presented in table 3) should be upper for running than for walking but this is not the case (for ACC ). Same for BPM.
Minor remarks:
line 241: Wasserstein distance is noted Wass and eq. 1" it is noted W.
line 111; replace chapter by Section. Same in the last paragraph of Section 1 (and throughout the paper).
Correct references [5] and [6]
How are computed the results presented in Table2? In this table, the results obtained with Dempster's rule should be given.
Table 5, insert square bracket around 14.
Reviewer 2 Report
1. Authors need to further increase the number of experiments
2. Observations and Justification for the results is inadequate
3. Comparision of results are presented without sound proof/justification.
4. Motivating example is still missing for the reader and also potential applications (real-world) and future scope of work.
Round 3
Reviewer 1 Report
I am still not convinced by the explanation given by the authors.
Comment on answer #1:
The change from 5 elements in the discernment framework to 2 does not change my feeling. Moreover, in this answer, the authors say: "There is no similar form of m(A, B) to BPA in the paper, for example, m({stand, walk}), because it is impossible for people to stand and walk at the same time."
In belief function theory, the hypotheses are exclusive (as stated in line 134 of the article). This means that only one of the assumptions in the discernment framework is the solution to the problem. For example with a framework \omega={stand, walk}, people either walk or they stand. Saying m({stand, walk}) does not mean that people are standing and at the same time that he is walking but simply that the solution is in the set {stand, walk} (we "hesitate" between the fact that he is walking or standing). The answer given by the authors makes me doubt their mastery of the theory of belief functions.
In the experiment n°2 (Table n°6), to calculate the accuracy is the number of data of each class the same because it seems to me that the accuracy given in the last column is the average of the accuracy. This value is obtained for how much data in database?
Finally, the comments on this experiment should be better organized. Currently, we talk about table 6, then figure 4, then the coefficient \kappa (with table 7), then we come back to figures 3 and 4. The authors should talk about Table 6, the coefficient \kappa, and finally Figures 3 and 4.